# A Maximum Entropy Production Hypothesis for Time Varying Climate Problems: Illustration on a Conceptual Model for the Seasonal Cycle

**DOI:** 10.3390/e22090966

**Published:** 2020-08-31

**Authors:** Vincent Labarre, Didier Paillard, Bérengère Dubrulle

**Affiliations:** 1Laboratoire des Sciences du Climat et de l’Environnement, UMR 8212 CEA-CNRS-UVSQ, IPSL and Université Paris-Saclay, 91191 Gif-sur-Yvette, France; 2SPEC, CEA, CNRS, Université Paris-Saclay, CEA Saclay, 91191 Gif-sur-Yvette, France; berengere.dubrulle@cea.fr

**Keywords:** maximum entropy production, energy balance model, closure hypothesis, seasonal cycle

## Abstract

We investigated the applicability of the maximum entropy production hypothesis to time-varying problems, in particular, the seasonal cycle using a conceptual model. Contrarily to existing models, only the advective part of the energy fluxes is optimized, while conductive energy fluxes that store energy in the ground are represented by a diffusive law. We observed that this distinction between energy fluxes allows for a more realistic response of the system. In particular, a lag is naturally observed for the ground temperature. This study therefore shows that not all energy fluxes should be optimized in energy balance models using the maximum entropy production hypothesis, but only the fast convective (turbulent) part.

## 1. Introduction

Time varying problems are ubiquitous in climate modeling. Indeed, the amount of radiative energy coming from the Sun, which is the dominant contribution to the energy budget of the Earth, depends on time. The variations of this radiative forcing occur on different time scales (from hours for the diurnal cycle to several thousand years for the variations of orbital parameters). To predict the time evolutions of relevant variables (temperature, humidity, wind, …) around the globe, one must consider how the different components of the climate system respond to this forcing, and how these components interact with each other.

The representation of the diurnal and seasonal variations of surface temperature in climate models depends on many processes. The type of surface (ocean or land with more or less vegetation) influences the sensible and latent energy fluxes between the ground and the atmosphere. For the ocean, the radiation that penetrates over a depth of ≃70 m under the surface allows the ocean to buffer the radiative energy input during daytime and more (days, weeks or months). This is not the case for land because of its opacity. In addition, change of surface type naturally modifies its heat capacity and the kind of energy transfers that can occur in the medium. For example, the heat capacity is more important for water than for ground, and convection usually does not occur in the ground. Overall, change in surface type has an important impact on the mass (water vapor and CO_2_), momentum (by friction) and energy (sensible and latent heat) exchanges between ground and atmosphere. Obviously, the water and energy budget at the surface, and so the representation of the turbulent sensible and latent heat fluxes, is of particular importance to describe these phenomena.

Most of global climate models use parameterizations to represent sub-grid processes like turbulent fluxes. For example, one of the most popular representations of surface fluxes relies on semi-empirical relations called aerodynamic bulk formulas. They are based on the application of the similitude theory on the atmospheric boundary layer [1]. Such relations aim to predict the relation between the turbulent fluxes between ground and atmosphere as a function of the wind and differences of humidity and temperature. The applicability of these formulas and the estimations of the empirical coefficients have been the subject of numerous studies [2,3,4,5,6]. The issues that are raised by the use of aerodynamic formulas are the same than for the closures used for the representations of turbulent fluxes in the atmosphere and ocean [7,8]. Namely, the applicability of these formulas to various conditions is still an open question. For example, the coefficients appearing in the simplest sensible heat flux parameterizations depend on the stability of the atmosphere. This issue has been addressed by using conditional formulas [9] or by adding more free parameters to fit experimental data over a wider range of conditions [10]. Global climate models are used to perform many experiments such as sensitivity of surface temperature tests [11], but usually fail to represent the last trends of the seasonal cycle [12].

The question of the phase and amplitude of the ground surface temperature during the seasonal (or diurnal) cycle can also be investigated with energy balance models. Such models can be useful to explore how local processes may cause variability in the phase and amplitude of the seasonal cycle of surface temperature [12]. For example, a model based only on a simplified energy budget and a maximum power closure hypothesis to fix the surface energy flux, has been constructed to explain the difference in sensitivity over ocean and land [13]. The maximum power closure hypothesis takes the form of a constrained variational problem that does not involve free parameters. It belongs to a class of thermodynamic optimality hypotheses used in various contexts [14]. Similarly, some climate models have been constructed on the maximum entropy production hypothesis or “principle” (hereafter MEP) [15,16,17,18] (see also the discussion [19] and the reviews [20,21]). All of them are based on the energy budget equation, and MEP appears as a closure hypothesis to determine the energy fluxes.

The statute of the MEP hypothesis is still not clear. It is argued in [22] that MEP is an inference. Other studies suggest that among several steady-states, the most stable is the one of maximum entropy production. This fact is observed for example in numerical simulation of the oceanic circulation [23] or simple bistable chemical system [24]. In spite of the lack of strong physical understanding, MEP models give correct estimations for the meridional atmospheric energy fluxes and temperature profiles in the stationary state [15].

In the context of time-varying problems, it has been observed in [25] that the MEP hypothesis could not reproduce a lag between the forcing and the response in the case of an antisymmetric forcing, which is quite unexpected for a system with thermal inertia. The discussion [26] suggested that there is a link between the maximum entropy production hypothesis and the minimum mixing time of a nonequilibrium system. If true, this statement implies that the MEP hypothesis should not be used to represent slow dynamics, but only the fast turbulent mixing.

Typically, the seasonal cycle is characterized by the amplitude and phase of the periodic surface temperature (relative to the periodic local solar insolation). The difference between the temperature and local insolation phases is called the lag, and the ratio of the amplitudes is the gain [12]. We propose a conceptual model of four boxes where the two upper boxes are forced radiatively, and energy is exchanged with a turbulent energy flux that is optimized with MEP. These two boxes are connected to buffers (bottom boxes) that roughly represent ground and store/release energy by conduction. Interestingly, the lag of the upper box can be modified by a parameter that represents the heating time of the upper box by diffusion. This behavior was not observed in [25]. It suggests that the MEP hypothesis could be used to construct simple climate models for time varying problems if we take care to separate slow processes (energy storage) and fast processes (turbulent mixing), and apply MEP only to the last ones. The outline of the paper is as follows:

In Section 2, we present the MEP hypothesis formalism in energy balance models, and we present our conceptual model with four boxes and introduce the control parameters. In Section 3, we study the influence of the control parameters on the temperatures and the energy fluxes represented with MEP. We discuss the results in Section 4.

## 2. Materials and Methods

### 2.1. Time Dependent MEP-Based Climate Models

A climate model is usually a discretization of atmosphere and ground (ocean or land) into boxes that exchange mass, momentum, and energy. Atmospheric models explicitly represent these exchanges, with parameterizations for the representation of the small scales’ motions. Energy balance models are only based on the energy budget equation. We represent the climate system by $\left(H_{a}+H_{g}+1\right)\times L_{x}\times L_{y}$ boxes where $H_{a}$ is the number of atmospheric boxes in each vertical column, $H_{g}$ is the number of ground boxes in each vertical column and $L_{x}\times L_{y}$ is the number of boxes in each horizontal layer. The position of a box is indexed by its discrete position $\left(x,y,z\right)\in \left[1:L_{x}\right]\times \left[1:L_{y}\right]\times \left[-H_{g}:H_{a}\right]$. The layer z=0 corresponds to the thin surface of the ground that is radiatively heated/cooled.

To construct an energy balance model, one must specify the energy content as a function of the thermodynamic variables (temperature, humidity, etc.) and properties of the media (heat capacities). We also have to represent the energy fluxes between different boxes (Figure 1). Those energy fluxes are usually mainly driven by turbulent (sub-grid) motions and are difficult to represent in practice. We note:Atmospheric (z>0) and oceanic (z≤0) energy fluxes $F_{xyz,x^{\prime }y^{\prime }z^{\prime }}$ from box (x,y,z) to $\left(x^{\prime },y^{\prime },z^{\prime }\right)$ (in *W*);Surface energy fluxes due to sensible and latent heat fluxes between the ground and the atmosphere $Q_{xy}$ (in *W*);Conduction in the ground between box (x,y,z) and box (x,y,z+1) is noted $D_{xyz}\left[T\right]$ (in *W*). In oceans, energy transfers can also occur by turbulent mixing and oceanic circulation. We have neglected the horizontal diffusion in the ground due to the large aspect ratio of the Earth’s crust.

It is also necessary to model the external radiative forcing $R_{xyz}\left[T\right]$ (in *W*) that is a functional of the temperature *T* and greenhouse gases concentrations. The radiative forcing is also strongly affected by the surface albedo which depends on the kind of surface (land or ocean, type of soil/vegetation, ice cover, …). Physically, $R_{xyz}\left[T\right]$ represents the net radiative balance of box (x,y,z) (in *W*). Boxes represent different kind of material, with a more or less important size. Therefore, their properties such as heat capacities $C_{xyz}$ (in J·$K^{-1}$) vary from a box to another.

Considering only the sensible heat term in the expression of the energy of each box, the energy budget can be written as follows ∀x,y: (1)$$\begin{array}{cc} \; & C_{xyz}{\dot{T}}_{xyz}=R_{xyz}\left[T\right]+\gamma_{xyz},\;z\in \left[2:H_{a}\right], \end{array}$$(2)$$\begin{array}{cc} \; & C_{xy1}{\dot{T}}_{xy1}=R_{xy1}\left[T\right]+\gamma_{xy1}+Q_{xy}, \end{array}$$(3)$$\begin{array}{cc} \; & C_{xy0}{\dot{T}}_{xy0}=R_{xy0}\left[T\right]-Q_{xy}+D_{xy-1}\left[T\right], \end{array}$$(4)$$\begin{array}{cc} \; & C_{xyz}{\dot{T}}_{xyz}=-D_{xyz}\left[T\right]+D_{xy(z-1)}\left[T\right],\;z\in \left[-1:-H_{g}\right], \end{array}$$
where we have introduced the convergence of atmospheric energy fluxes $\gamma_{xyz}=\sum_{x^{\prime }y^{\prime }z^{\prime }}\;\left(F_{x^{\prime }y^{\prime }z^{\prime },xyz}-F_{xyz,x^{\prime }y^{\prime }z^{\prime }}\right)$. In usual energy balance models, $F_{xyz,x^{\prime }y^{\prime }z^{\prime }}$ and $Q_{xy}$ are related to the gradient of temperature and humidity by parameterizations (sometimes only by diffusion, as in references [27,28,29,30,31]). Alternatively, the maximum entropy production hypothesis consists of determining $\gamma_{xyz}$ and $Q_{xy}$ by maximizing the entropy production created by the turbulent sensible heat exchanges over all the domain and time t∈[0:T]:(5)$$\sigma =\int_{0}^{T}\;\left\{\underset{\mathrm{upper}\;\mathrm{atmosphere}}{\underbrace{\sum_{x,y,z\in [2:H_{a}]}\;\frac{\gamma_{xyz}}{T_{xyz}}}}+\underset{\mathrm{boundary}\;\mathrm{layer}}{\underbrace{\sum_{x,y}\;\frac{\gamma_{xy1}+Q_{xy}}{T_{xy1}}}}-\underset{\mathrm{surface}}{\underbrace{\sum_{x,y}\;\frac{Q_{xy}}{T_{xy0}}}}\right\}dt,$$
taking into account that the energy fluxes *F* are, by definition, internal to the atmosphere so the sum of its convergence over the domain is zero at any time:(6)$$\sum_{x,y,z}\;\gamma_{xyz}=0\;\forall t\in \left[0:T\right].$$

Maximum entropy production is here applied to the convergence of turbulent energy fluxes and we directly compute the temperatures. Physically, it represents an efficient heat mixing that is only constrained by the energy budget (Equations (Equation 1)–(4)) and the constraint (Equation (Equation 6)). The reason why we do not optimize the entropy production created inside the ground layer is that the heat exchange is not ensured by the fast convective processes but by slow conductive processes. For the same reason, only the variables directly involved in the turbulent entropy production are considered as variables of the optimization problem. Following [25], the solution of the problem is a critical point of the following functional:(7)$$A\left[\gamma_{xyz},Q_{xy},\beta \right]=\int_{0}^{T}\;\left\{\sum_{x,y,z\in [2:H_{a}]}\;\frac{\gamma_{xyz}}{T_{xyz}}+\sum_{x,y}\;\frac{\gamma_{xy1}+Q_{xy}}{T_{xy1}}-\sum_{x,y}\;\frac{Q_{xy}}{T_{xy0}}-\beta \sum_{x,y,z\in [1:H_{a}]}\;\gamma_{xyz}\right\}dt,$$
where β is the function of time that is the continuous set of Lagrange multipliers associated with the constraints (Equation 6) for all times. We express this functional in terms of the temperatures using the energy budget Equations (Equation 1)–(4) as follows:(8)$$\begin{array}{cc} \; & A\left[T_{xyz},{\dot{T}}_{xyz},\beta \right]=\int_{0}^{T}\;L\left[T_{xyz},{\dot{T}}_{xyz},\beta \right]dt,\\ \; & \mathrm{where} \end{array}$$(9)$$\begin{array}{cc} L[T_{xyz},{\dot{T}}_{xyz},\beta ] & =\sum_{x,y,z\in [1:H_{a}]}\;\left(C_{xyz}{\dot{T}}_{xyz}-R_{xyz}\left[T\right]\right)\left(\frac{1}{T_{xyz}}-\beta \right)\\ \; & +\sum_{x,y}\;\left(C_{xy0}{\dot{T}}_{xy0}-R_{xy0}\left[T\right]-D_{xy-1}\left[T\right]\right)\left(\frac{1}{T_{xy0}}-\beta \right) \end{array}$$
is the Lagrangian of the optimization problem. It can be expressed in a more compact form
(10)$$\begin{array}{cc} \; & L\left[T_{xyz},{\dot{T}}_{xyz},\beta \right]=\sum_{x,y,z\in [0:H_{a}]}\;\left(C_{xyz}{\dot{T}}_{xyz}-F_{xyz}\left[T\right]\right)\left(\frac{1}{T_{xyz}}-\beta \right)\\ \; & \mathrm{where} \end{array}$$
(11)$$\begin{array}{cc} \; & F_{xyz}\left[T\right]=R_{xyz}\left[T\right]+D_{xy(z-1)}\left[T\right]-D_{xyz}\left[T\right] \end{array}$$
simply represents the convergence of energy fluxes that are not optimized with MEP, namely, the sum of the net radiative energy input and the convergence of conductive energy fluxes. The diffusive part of the energy transport in the atmosphere is negligible compared to the convective part so we consider that $D_{xyz}=0\;\forall x,y,z\geq 0$. We observe that the sum in the Lagrangian is performed over all boxes where there is turbulent energy fluxes (optimized with MEP). At this point, it is worth mentioning that MEP makes no distinction on the nature of the energy fluxes that are optimized (surface or atmospheric heat fluxes, sensible or latent, diffusive or convective). The solution of the optimization problem satisfies the following Euler–Lagrange equations (see, for example, [32] for an introduction to variational problems in physics):(12)$$\begin{array}{cc} \; & \frac{\partial L}{\partial T_{xyz}}-\frac{d}{dt}\left[\frac{\partial L}{\partial {\dot{T}}_{xyz}}\right]=0\;\forall x,y,z\in \left[0:H_{a}\right] \end{array}$$(13)$$\begin{array}{cc} \; & \frac{\partial L}{\partial \beta }=0. \end{array}$$

These equations have to be complemented by the energy budget equations in the ground for z<0. All in all, we then obtain the following set of equations:(14)$$\begin{array}{cc} \; & C_{xyz}\dot{\beta }+\left(\sum_{x^{\prime }y^{\prime }z^{\prime }\in \left[0,H_{a}\right]}\;\frac{\partial F_{x^{\prime }y^{\prime }z^{\prime }}\left[T\right]}{\partial T_{xyz}}\right)\beta +\\ \; & \frac{F_{xyz}\left[T\right]}{T_{xyz}^{2}}-\sum_{x^{\prime }y^{\prime }z^{\prime }\in \left[0,H_{a}\right]}\;\left(\frac{\partial F_{x^{\prime }y^{\prime }z^{\prime }}\left[T\right]}{\partial T_{xyz}}\frac{1}{T_{x^{\prime }y^{\prime }z^{\prime }}}\right)=0\;\forall x,y,z\in \left[0:H_{a}\right], \end{array}$$(15)$$\begin{array}{cc} \; & \sum_{x,y,z\in [0:H_{a}]}\;\left(C_{xyz}{\dot{T}}_{xyz}-F_{xyz}\left[T\right]\right)=0, \end{array}$$(16)$$\begin{array}{cc} \; & C_{xyz}{\dot{T}}_{xyz}=-D_{xyz}\left[T\right]+D_{xy(z-1)}\left[T\right],\;\mathrm{for}\;z\in \left[-H_{g}:-1\right]. \end{array}$$

The only inputs of such a model are the discretization of the domain, the radiative model $R_{xyz}\left[T\right]$ and the representation of the non-turbulent (slow) heat fluxes (here the conduction in the ground). The differences with stationary MEP are that the radiative forcing depends on time and also that the slow variables (here the ground temperatures) can influence the fast variables through (16).

One notes that it is tempting to write the continuous field theory associated with the MEP hypothesis. Is this case, spatial derivatives of the temperature field can appear in the Lagrangian (i.e., $L[T_{xyz},{\dot{T}}_{xyz},\beta ]$ is replaced $L[T,\partial_{\mu }T,\beta ]$ where μ=(t,x,y,z)), so the discrete sums over boxes should be replaced by an integration, and Equations (Equation 14)–(16) should be replaced by partial differential equations (PDEs). However, contrary to systems usually described by classical field theories (like elasticity or classical electromagnetism), the system is divided here into several sub-systems (atmosphere, surface, ground) which makes it necessary to use finite difference at least to represent the interfaces between sub-systems (the surface energy flux $Q_{xy}$ for example). Thus, expressing the optimization with discrete space coordinates (i.e., dividing the system into boxes) is more pragmatic, even if less elegant. For more, in the end, the climate modeler will need to discretize the equation to solve it numerically, at least for a sufficiently realistic radiative model R[T]. Thus, the derivation used here is more direct when we are interested in the applications to climate modeling, even if it is less usual when one compares it to the field’s usual formalism.

### 2.2. Simplified Case

To analyze the physical relevance of MEP hypothesis and get insights into the effect it has on the represented variables, it is instructive to consider the model with only four boxes shown in Figure 2. It represents two columns of two layers. The upper boxes represent the part of the system that is forced radiatively (typically the troposphere and the thin ground surface) by source terms $R_{i},\;i=1,2$. They can exchange energy by a “turbulent” heat flux $F=\gamma_{2}=-\gamma_{1}$ that will be optimized with MEP. We note their temperature $T_{ui},\;i=1,2$. The two bottom boxes represent buffers of energy that store/release energy by conduction $D_{i},\;i=1,2$ to the upper boxes (typically the ground). We note their temperature $T_{bi},\;i=1,2$. $R_{i}$, *F* and $D_{i}$ are expressed in *W*. We consider a local radiative forcing (i.e., that depends only on the box temperature $R_{i}\left(T_{ui}\right)$ and external parameters), which is, of course, over-simplistic for a true atmospheric column, but allow for useful simplifications. Furthermore, one assumes that the radiative forcing can be linearized around a temperature $T_{0i}$ that physically represents the radiative equilibrium solution of the problem (i.e., the temperature of the upper box $T_{ui}$ in the absence of thermal inertia, no convective fluxes $\gamma_{i}=0$ and no diffusive heat fluxes $D_{i}=0$). The linearization around $T_{0i}$ assumes that the temperatures stay close to the radiative equilibrium solution. In this toy model, the atmosphere is reduced to only one layer (that even includes the surface layer). However, for models with several atmospheric layers and a realistic description of the radiative forcing, the radiative equilibrium temperatures are not the same for all the boxes. The radiative forcing then reads
(17)$$R_{i}=r_{i}\left(T_{0i}-T_{ui}\right),$$
where $r_{i}$ are extensive quantities (in *W*·$K^{-1}$) proportional to the size of the box *i*. Conduction is represented by
(18)$$D_{i}=k_{i}\left(T_{bi}-T_{ui}\right),$$
where $k_{i}$ is also an extensive quantity (in *W*·$K^{-1}$). We consider the homogeneous case where $r_{i}=r$ and $k_{i}=k$ for i=1,2. It can be shown that the Equations (Equation 14)–(16) reduce to
(19)$$\begin{array}{cc} \; & C_{u}\dot{\beta }-\left(r+k\right)\beta -\frac{rT_{0i}+kT_{bi}}{T_{ui}^{2}}=0\;i=1,2, \end{array}$$
(20)$$\begin{array}{cc} \; & \sum_{i=1}^{2}\;\left\{C_{u}{\dot{T}}_{ui}-r\left(T_{0i}-T_{ui}\right)-k\left(T_{bi}-T_{ui}\right)\right\}=0, \end{array}$$
(21)$$\begin{array}{cc} \; & C_{b}{\dot{T}}_{bi}=-k\left(T_{bi}-T_{ui}\right)\;i=1,2. \end{array}$$

The case k=0 corresponds to the model presented in [25]. In this case, one has
(22)$$T_{ui}=\sqrt{\frac{rT_{0i}}{C_{u}\dot{\beta }-r\beta }}\equiv \alpha \left(t\right)\sqrt{T_{0i}},$$
such that $T_{ui}$ is directly linked to the forcing $T_{0i}$, modulated by the function $\alpha \left(t\right)\equiv \sqrt{r}/\sqrt{C_{u}\dot{\beta }-r\beta }$. This solution does not correspond to a “true” local lag due to the local energy storage because $\sqrt{T_{0i}}$ is naturally in phase with the local forcing, and α(t) does not depend on *i* so it represents a global response (independent of the position) of the system at time *t*. In contrast, in the case k≠0, we have
(23)$$T_{ui}=\sqrt{\frac{rT_{0i}+kT_{bi}}{C_{u}\dot{\beta }-r\beta }}\equiv \alpha \left(t\right)\sqrt{T_{0i}+\frac{k}{r}T_{bi}}.$$

The response is still non-local due to the use of MEP and one still have the global modulation α(t). Yet, we naturally have introduced a “true” lag because $T_{bi}$ is related to $T_{ui}$ through the diffusion Equation (Equation 19). We expect that $T_{ui}$ will be in phase with the forcing temperature $T_{0i}$ if k≪r. On the contrary, $T_{ui}$ will be in phase with the buffer temperature $T_{bi}$ if r≪k. For real climatic conditions, we obviously observe an intermediate situation.

To illustrate this point, we show the temperatures and the energy flux during the cycle for a typical case when k≪r and with various values of $C_{u}$ in Figure 3. We observed that there was almost no lag of the upper boxes temperatures, even if we increased $C_{u}$ by several orders of magnitude as in [25]. That is quite unexpected in a real system since the heat capacity naturally controls the thermal inertia of a system. It is explained here by the fact that the energy flux *F* which maximizes the entropy production adjusts with the change of the heat capacity such that we do not observe a lag. It means that the energy storage (and hence the induced lag) cannot be represented with the MEP hypothesis alone.

If the geometry (i.e., the sizes of the boxes) is fixed, the problem depends on the heat capacities $C_{u}$ and $C_{b}$, and the conductivity *k*. In the context of a cyclic forcing, the radiative forcing is characterized by a typical time τ, a reference temperature $\langle T_{0}\rangle$, an amplitude $\Delta T_{0}$ and the constant *r*. These seven physical quantities depend on four physical units (mass, time, length, temperature). Through dimensional analysis, we can characterize the problem through only three dimensionless quantities. We chose the following dimensionless numbers
(24)$$N_{b}=\frac{C_{b}}{k\tau },\;N_{r}=\frac{C_{u}}{r\tau }\;\mathrm{and}\;N_{k}=\frac{C_{u}}{k\tau },$$
which respectively represent the dimensionless (divided by the cycle’s time τ) inertial thermal time scale by conduction in the bottom box, the inertial thermal time scale by radiative forcing in the upper box and the inertial thermal time scale by conduction in the upper box. These parameters control the relative amplitude and lag between temperatures. $\tau N_{b}$ is the typical lag of the ground (≃ two weeks to two months). $\tau N_{r}$ is the typical lag of the atmospheric response and the radiative forcing (≪1 year). $N_{k}$ controls the influence of the bottom on the top by diffusion. Then, the evolution equations for our model can be written as
(25)$$\begin{array}{cc} \; & \dot{\beta }-\left(\frac{1}{N_{r}}+\frac{1}{N_{k}}\right)\beta -\frac{\frac{1}{N_{r}}T_{0i}+\frac{1}{N_{k}}T_{bi}}{T_{ui}^{2}}=0\;i=1,2, \end{array}$$
(26)$$\begin{array}{cc} \; & \sum_{i=1}^{2}\;\left\{{\dot{T}}_{ui}-\frac{1}{N_{r}}\left(T_{0i}-T_{ui}\right)-\frac{1}{N_{k}}\left(T_{bi}-T_{ui}\right)\right\}=0, \end{array}$$
(27)$$\begin{array}{cc} \; & {\dot{T}}_{bi}=-\frac{1}{N_{b}}\left(T_{bi}-T_{ui}\right)\;i=1,2. \end{array}$$
where the time has been scaled by τ. The temperatures have not been scaled (i.e., they are still expressed in *K*) for simplicity. This set of equation has no explicit solution but it can be solved numerically using the Newton’s method (Appendix B).

We studied the evolution of the temperatures’ gains and lags with the variations of the control parameters $N_{b}$, $N_{r}$ and $N_{k}$. These quantities depend mainly on the heat capacities (and so the dimensionless parameters introduced above). It is also useful to track the evolution of the energy flux between the upper boxes. For this, we introduce the quantity
(28)$$q=\frac{\tau F}{C_{u}}=\frac{1}{N_{r}}\left(T_{01}-T_{u1}\right)+\frac{1}{N_{k}}\left(T_{b1}-T_{u1}\right)-{\dot{T}}_{u1},$$
that can be tracked as a diagnostic variable. It represents the temperature elevation of an upper box due to an energy input τF.

## 3. Results

We focused on the case of an antisymmetric forcing: $T_{01}=300+10\sin\left(\frac{2\pi }{\tau }t\right)\;K$ and $T_{02}=300-10\sin\left(\frac{2\pi }{\tau }t\right)\;K$, and varied the control parameters $N_{b}$, $N_{r}$ and $N_{k}$ to study their influences. In this case, the two columns can be viewed as a crude representation of the two hemispheres of the Earth. Due to the symmetry, we only present the temperatures of the first column $T_{u1}$ and $T_{b1}$. Indeed, $T_{u2}$ and $T_{b2}$ are the antisymmetric versions of $T_{u1}$ and $T_{b1}$ with respect to the line T=300K.

Influence of $N_{b}$: lag of ground:

We clearly observe in Figure 4a that $N_{b}$ controls the lag and amplitude of the bottom temperature (with respect to the upper box). This parameter does not significantly affect the upper box temperature (Figure 4a,b) nor *q*. As $N_{b}$ increases, the amplitude of $T_{b}$ decreases and its lag increases. This is a classical behavior for diffusion. When $N_{b}$ is finite (i.e., k≠0), a local lag is naturally introduced, giving a more satisfactory temperature response than in [25]. *q* is not influenced by $N_{b}$.

Influence of $N_{k}$: bottom to top diffusion:

On Figure 5a, we observe that $N_{k}$ influences the lag and amplitude of the upper and bottom boxes temperatures (with respect to the radiative forcing). As $N_{k}$ increases, the lag decreases and the temperature amplitudes increase. That is easily understood since a high value for $N_{k}$ represents a large heating time of the upper boxes by diffusion, so they are mainly following the evolution of the radiative forcing. This parameter therefore controls the influence of the bottom ("buffer") on the upper box by conduction. As explained above, it induces a "true" lag of the upper boxes. *q* decreases with $N_{k}$ (Figure 5b).

Influence of $N_{r}$:

Results presented in Figure 6a show that $N_{r}$ controls the lag and amplitude of the upper boxes’ temperatures (with respect to the radiative forcing). This parameter has an impact on *q* (Figure 6b). As $N_{r}$ increases, the amplitude and the lag of *q* increase.

We observed that the lag of *q* is not the same as the lag of the upper boxes’ temperatures (in other words, there is a lag between the upper boxes’ temperatures and the energy flux). As a consequence, the local entropy production $F(1/T_{u1}-1/T_{u2})$ can be negative during the cycle. It is important to note that it does not necessarily contradict the 2nd law of nonequilibrium thermodynamics which says that the local entropy production due to diffusive heat exchanges is positive, since the definition of entropy production used here encompasses the convective fluxes. We now consider the case of an asymmetric forcing $T_{01}=290+10\sin\left(\frac{2\pi }{\tau }t\right)\;K$ and $T_{02}=310+3\sin\left(\frac{2\pi }{\tau }t+\frac{\pi }{4}\right)\;K$ with $N_{b}=0.1$, $N_{r}=0.001$ and $N_{k}=0.1$. The temperatures and *q* are shown on Figure 7a,b. The expected lag between the upper boxes and bottom boxes is of order $N_{b}$.

## 4. Discussion

Our model based on the MEP hypothesis has an over-simplistic representation of the Earth system. It was constructed to test the effects of the inclusion of energy storage by conduction in such models, in particular on the lag in the temperature response. The only aspect that MEP keeps about the dynamics is its ability to mix heat. Dynamical constraints have to be combined with the MEP hypothesis (see, for example, [33]) to obtain more realistic results. As discussed in [19], the relevant constraints depend on the problem, and MEP then appears as a useful closure hypothesis that fixes the remaining unknowns, namely, turbulent convective heat fluxes.

When MEP is used, we loose the precise nature and timescales of the different kinds of energy fluxes that are optimized. This means that MEP must obviously be used with caution. This study suggests that we need to have a separation of scales between the global mixing time due to fast turbulent processes, and the scale of interest (which is the period of the cycle here) in order to apply MEP to time-varying problems. This separation of scale is more or less valid for the atmospheric turbulent mixing, but is not true for the energy storage in the ground by conduction that operates on larger timescales. One notes that the separation into fast processes and slow processes recalls the quasi-static approximation, and that this separation depends on the problem considered.

An important question is why entropy production is the relevant quantity to optimize (and why not the power?). This study does not provide a definitive answer. However, one can give the following heuristic explanation: The maximum power hypothesis is interesting for “smart” systems like living organisms (or devices) that are adapting themselves (or adapted) in order to maximize their performances (for example, the energy consumption) despite a reduced efficiency. In such systems, work (the output power) is extracted from the system. On the contrary, convection is not a “smart” adaptative (or adapted) process, but mixes energy (and other quantities) inside the atmosphere so the entropy production is certainly the most pertinent quantity to optimize in this case.

As a final remark, one notes that the maximization of the global entropy production due to turbulent convection implicitly assumes that the turbulent convection is fast enough to mix the atmosphere globally so the global entropy production can be optimized.

We therefore think that it is possible to use MEP to represent turbulent atmospheric motions in more realistic energy balance models (i.e., with a more precise description of the climatic system and its associate radiative budget) to study the seasonal cycle.

## 5. Conclusions

We have used the MEP hypothesis in the context of a time-varying problem for a conceptual model. Contrarily to the existing MEP based models, only some energy fluxes were optimized with MEP (the convective ones) and energy was buffered via conduction. This feature naturally led to a lag in the local temperature response, even in the case of an antisymmetric forcing, where no significant lag was observed if we optimized all energy fluxes with MEP. This result suggests that MEP could be used to construct simple climate models for time-varying problems if slow processes, such as conduction in the ground, are decoupled from fast (convective) processes that are fixed by the optimization of entropy production.

## Figures and Tables

**Figure 1 entropy-22-00966-f001:**
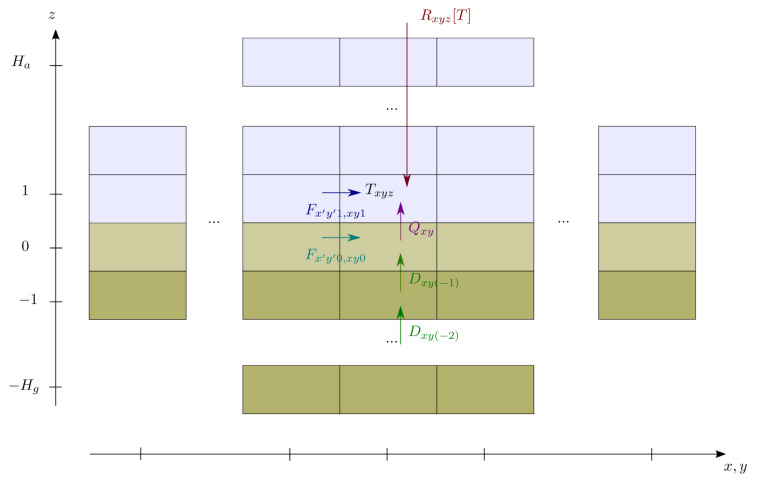
Schematic view of an energy balance model.

**Figure 2 entropy-22-00966-f002:**
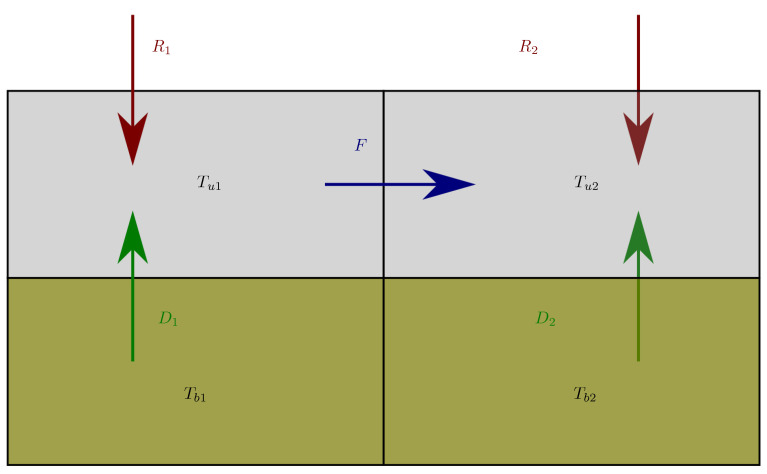
The four-box model.

**Figure 3 entropy-22-00966-f003:**
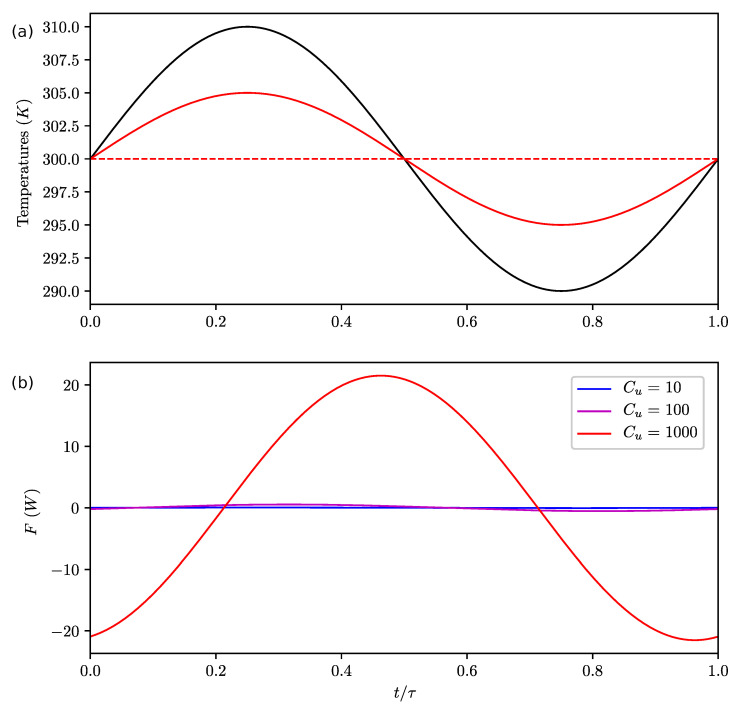
Evolution of temperatures in the first column $T_{u1}$ (full lines) and $T_{b1}$ (dashed lines); (**a**) and energy flux *F* (**b**) during the cycle for $C_{b}=1000,\tau =1000,k=0.001\ll r=1.5$, various values of $C_{u}$ and antisymmetric forcing temperatures: $T_{01}=300+10\sin\left(\frac{2\pi }{\tau }t\right)\;K$ and $T_{02}=300-10\sin\left(\frac{2\pi }{\tau }t\right)\;K$. The forcing temperature $T_{01}$ is shown in black. The temperatures in the second column $T_{u2}$, $T_{b2}$ and $T_{02}$ are the antisymmetric versions of $T_{u1}$, $T_{b1}$ and $T_{01}$ with respect to the line T=300K. We see that $C_{u}$ does not influence the temperatures. Here, there is no lag between the forcing and the temperature response.

**Figure 4 entropy-22-00966-f004:**
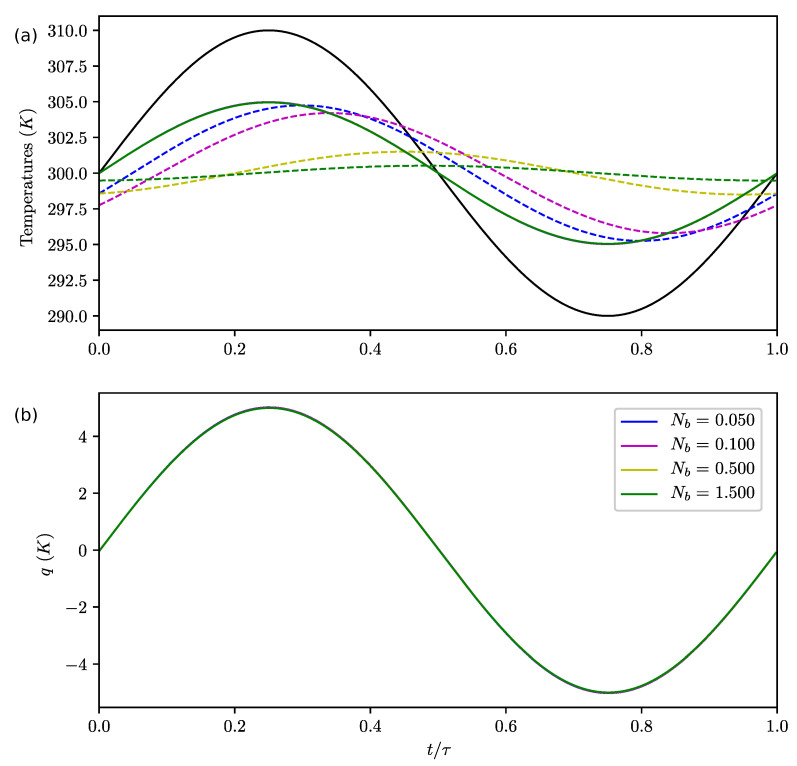
Evolutions of temperatures in the first column $T_{u1}$ (full lines) and $T_{b1}$ (dashed lines); (**a**) and *q* (**b**) during the cycle for varying $N_{b}$, $N_{r}=0.001$, $N_{k}=0.1$ and antisymmetric forcing. The forcing temperature $T_{01}$ is shown in black. The temperatures in the second column $T_{u2}$, $T_{b2}$ and $T_{02}$ are the antisymmetrics of $T_{u1}$, $T_{b1}$ and $T_{01}$ with respect to the line T=300K. The curves q(t) for different $N_{b}$ cannot be distinguished.

**Figure 5 entropy-22-00966-f005:**
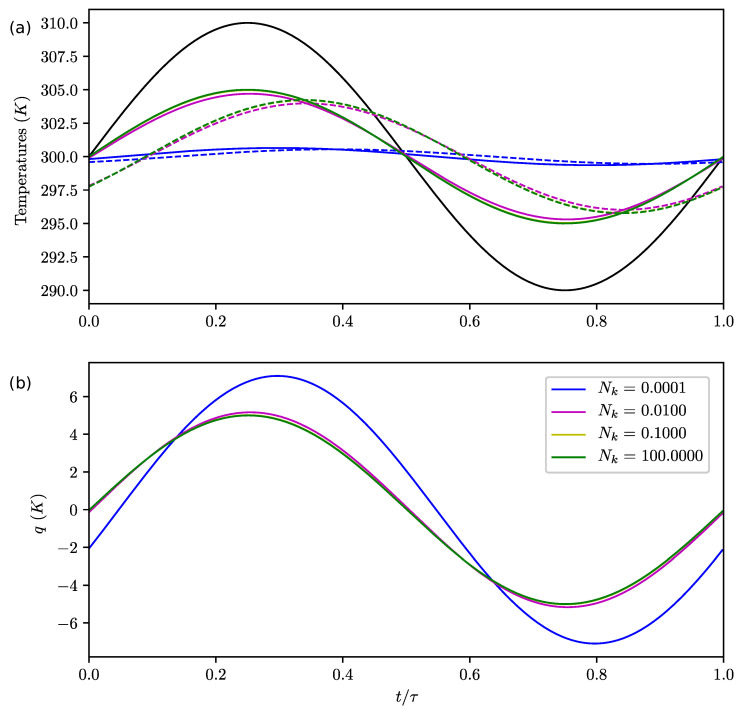
Same as Figure 4 for $N_{b}=0.1$, $N_{r}=0.001$ and various $N_{k}$.

**Figure 6 entropy-22-00966-f006:**
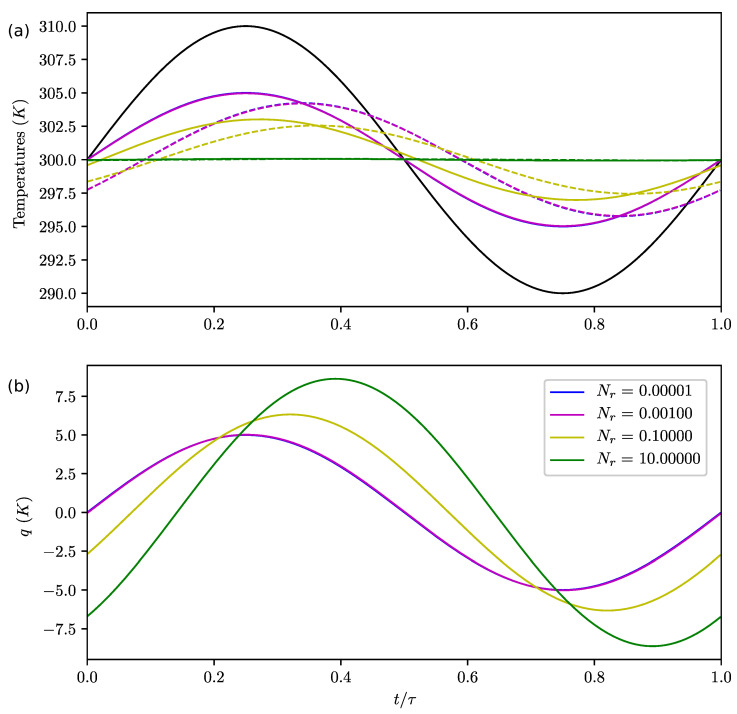
Same as Figure 4 for $N_{b}=0.1$, various $N_{r}$ and $N_{k}=0.1$.

**Figure 7 entropy-22-00966-f007:**
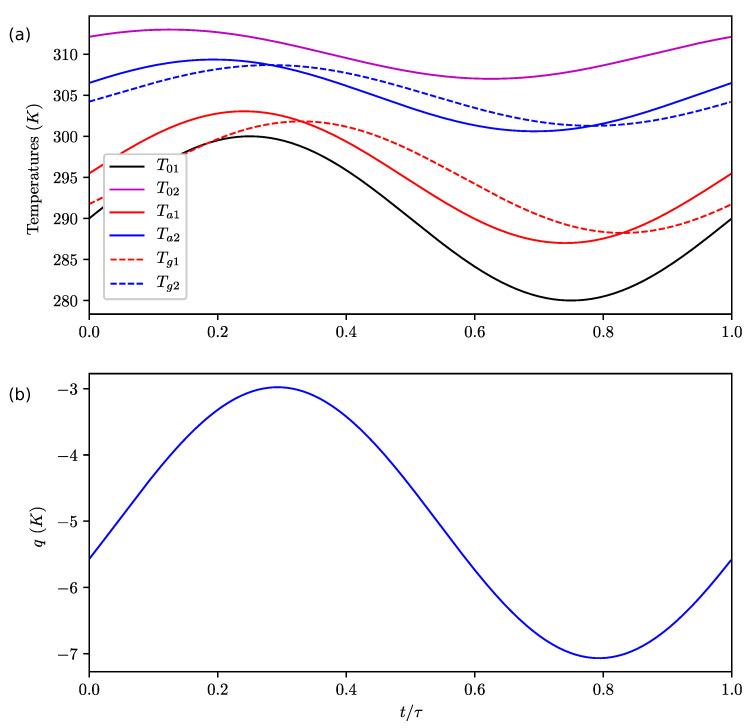
Evolution of temperatures (**a**) and *q* (**b**) during the cycle for $N_{b}=0.1$, $N_{r}=0.001$, $N_{k}=0.1$, $T_{01}=290+10\sin\left(\frac{2\pi }{\tau }t\right)\;K$ and $T_{02}=310+3\sin\left(\frac{2\pi }{\tau }t+\frac{\pi }{4}\right)\;K$.

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
