# Peer review of "A Maximum Entropy Production Hypothesis for Time Varying Climate Problems: Illustration on a Conceptual Model for the Seasonal Cycle"

_entropy, 2020, doi:10.3390/e22090966_

Round 1

Reviewer 1 Report

In this paper, the authors use the MEP hypothesis to model seasonal cycles. Their key observation is that not all modes of heat transfer should be optimized under the MEP hypothesis. They further show that only optimizing the advective mode of heat transfer gives realistic estimates. 

In lines 63-68 the authors present an interesting observation. It is indeed true that MEP advocates for the fastest pathway to a non-equilibrium state but whether MEP 'should not' be used for slow dynamics is open for debate. If we consider the log ratios of the path probabilities between two states, then the fraction is proportional to the entropy production (or rate of entropy production). Here the complexity of the end states are of importance as well. For example, a non-turbulent Rayleigh-Bénard convection beyond the critical Rayleigh number shows significant complexity as compared to the ground-state (the dynamics is slow). The reduction in entropy production due to this emergent complexity is offset by very high entropy production. Thus, MEP stays valid (see, sci. rep. (9) 10615, 2019). Similarly, in a turbulent Rayleigh-Bénard convection we have very high entropy production due to turbulent mixing which also leads to more complexity and thus the MEP is valid again (the dynamics in this case is fast). So, whether a process can be explained by the MEP largely depends upon the relative character of the initial and final states, i.e., whether they are high entropy producing - high complexity states or low entropy producing - low complexity states (see, sci. rep. (7) 14437, 2017). Maybe, the authors should elaborate on this aspect?

One question regarding their model in Eqn.8: this is the standard action equation from mechanics. I can understand that the authors have transformed the coordinates from (x,\dot{x},t) to (T, \dot{T},\beta). I however do not understand why it shouldn't be rather (T,\grad_\mu{T},\beta) where \mu = (t,x) like it is usually done in classical field theories. A little bit of insight here might help the reader. 

Towards the end the authors do describe the limitation of their 'over-simplistic' model - the key aspect here is that their model works more or less well for the atmosphere with low heat capacity and fast turbulent mixing, but not for the case where the heat capacity is larger, or where the mode of heat transport is governed by conduction. So, what does this mean to a general audience? If, I want to apply this model what would be an ideal relatable situation where I can use this model, for example, a relatable atmospheric phenomenon (tornado?) or a relatable atmosphere type (thin layer?)?

Please check the references mentioned in text before Eqn.5: they don't seem to appear properly. 

Reviewer 2 Report

It is an interesting contribution to the Journal that must be publshed. Well developed and detailed approach. The definition of the problem is well identified. However, there are a pair of suggestions and questions that should be addressed before publication:

1. It is recommended to write in third person.

2. The authors in Sub-section 2.1 comment that the "convergence gxyz are internal to the atmosphere and must sum up to zero at any time". Explain in detail this assumption.

3. Please check the references: [27? ? –30]

4. Could the authors provide examples of components of the climate system where heat capacity is larger? and others where the heat transport is governed by conduction?. As a suggetsion in the discussion section.

Reviewer 3 Report

I have read the manuscript “Maximum Entropy Production Hypothesis for Time Varying Climate Problems: Illustration on a Conceptual Model for the Seasonal Cycle” by Vincent Labarre et al.

The paper is sufficiently clear and as far as I could check the mathematical development seems correct, I just have some minor questions and concerns that I think the authors will be able successfully overcome.

  1. Above Eq. (5) are Eqs. (28)-(29) missing?
  2. Are Eqs. (1)-(4) Energy “flux” budget? since they are expressed in J/s.
  3. At the beginning of page 5: What do you mean with “a continuous extension of a Lagrange multiplier”? is it not the usual Lagrange multiplier?
  4. Before Eq. (17) could you say a few more words about the temperatures T_0i, what does the linearization of the radiative forcing mean? Does the equilibrium solution imply the same temperature for all cells in a column?
  5. Regarding Figure 3: T01 and T02 have, according to the caption, a phase shift of Pi, why are these temperature profiles so different? Additionally, are the T0i profiles assumed?
  6. At the end of page 7: “The first is the fact that it was shown that MEP does not lead to physically consistent results if Nr is not small (i.e. if the time scale of variation of the forcing is not large compared to the heating time of the medium).” To what physical inconsistencies do this phrase refer? And could you explain a little more about the “time scale of variation of the forcing” as well as the “heating time of the medium”
  7. A lag is defined just before Eq. (28) but it is used in other places before, for example, where the function alpha is introduced, please define it the first time it is used in the text.
  8. In line 163 a local negative entropy is discussed, however, sigma (if you should keep it, please use another sigma, since in Eq. (5) it refers only to the fast dynamics component) is that of an open system (due to the external radiative forcing) and there is no contradiction with the 2nd law, then, this discussion is not needed.
  9. Please make T01 and T02 more distinguishable in Fig. 7a.
  10. Additionally, is the heat associated to phase transitions (evaporation and rain) considered? Clouds sizes and rain are considered in some cases as a SOC (self-organized criticality) phenomenon, and scale free, then, it could be expected to contribute in slow and fast dynamics. If this contribution is already accounted in the phenomenological model maybe a comment can be useful.
  11. As a final comment, I would like to see some more specific comments/discussion regarding the results liked to maximum entropy production, in Eqs. (22) and (23), and also in Figs. 4-7. What contribution are obtained that reinforce the idea that MEP hypothesis is relevant only for fast dynamics and why no other function, such a power (to name one). I think this point should be strengthened since it is in the core of the paper.

I find the paper interesting since it offers a quite simple but significative basic description of a complex dynamics based on thermodynamic constraints. This could give some theoretical insight regarding the mechanisms "chosen" in nature and possible optimal mechanisms inside complex phenomena.

If the authors answer my questions, I would recommend the paper for publication in Entropy.

Round 2

Reviewer 1 Report

The authors have made the suggested changes where ever necessary and have also responded to my comments satisfactorily. I, therefore recommend publication. 

Reviewer 3 Report

I have read the revised version of the present manuscript. The Authors have answered all my questions, then, I recommend its publication in Entropy.